# Preparing and Wear-Resisting Property of Al_2_O_3_/Cu Composite Material Enhanced Using Novel In Situ Generated Al_2_O_3_ Nanoparticles

**DOI:** 10.3390/ma16134819

**Published:** 2023-07-04

**Authors:** Youming Chen, Rafi Ud-din, Teng Yang, Tao Li, Chuanghao Li, Aimin Chu, Yuping Zhao

**Affiliations:** 1Hunan Provincial Key Defense Laboratory of High Temperature Wear-Resisting Materials and Preparation Technology, School of Materials Science and Engineering, Hunan University of Science and Technology, Xiangtan 411201, China; ymchen@hnust.edu.cn (Y.C.);; 2Materials Division, PINSTECH, Post Office Nilore, Islamabad 44000, Pakistan; 3School of Civil and Engineering, Hunan University of Science and Technology, Xiangtan 411201, China

**Keywords:** metals, composite material, mechanical properties, X-ray diffraction, microstructure

## Abstract

Al_2_O_3_/Cu composite material (ACCM) are highly suitable for various advanced applications owing to its excellent properties. In the present work, a combination of the solution combustion synthesis and hydrogen reduction method was first employed to prepare Al_2_O_3_/Cu composite powder (ACCP), and subsequently ACCM was prepared by employing spark plasma sintering (SPS) technique. The effect of Al_2_O_3_ contents and SPS temperatures on the properties (relative density, hardness, friction coefficient, and electrical conductivity, et al.) of ACCM were investigated in detail. The results indicated that ACCM was very dense, and microstructure was consisted of fine Al_2_O_3_ particles evenly distributed in the Cu matrix. With the increase of SPS temperature, the relative density and hardness of ACCM had first increased and then decreased. At 775 °C, the relative density and hardness had attained the maximum values of 98.19% and 121.4 HV, respectively. With the increase of Al_2_O_3_ content, although the relative density of ACCM had gradually decreased, nevertheless, its friction coefficient had increased. Moreover, with the increase of Al_2_O_3_ contents, the hardness of ACCM first increased and then decreased, and reached the maximum value (121.4 HV) with 3 wt.% addition. On the contrary, the wear rate of ACCM had first decreased and then increased with the increase of Al_2_O_3_ contents, and attained the minimum (2.32 × 10^−5^ mm^3^/(N.m)) with 3 wt.% addition.

## 1. Introduction

During recent years, Cu matrix composites (CuMC) are found to be highly suitable for various advanced applications such as integrated circuit wire frame and spot welding electrode owing to its high thermal and electrical conductivity, superior mechanical properties, and excellent wear resistance [1,2,3]. Scholars, all over the world, have achieved significant results by investigating various aspects of CuMC [4,5,6]. CuMC can be divided into three types such as fiber reinforced, whisker reinforced, and particle reinforced. Among these three composite materials, the CuMC using particle-reinforced have found broader applications owing to its better isotropic properties, simpler preparation processes, and good processing properties. Common reinforcing particles include oxide ceramics (Al_2_O_3,_ ZrO_2_, SiO_2_ [7,8,9], etc.) and non-oxide ceramics (AlN, SiC, TiB_2_ [10,11,12], etc.). Out of these reinforcing, Al_2_O_3_ with its superior properties can significantly improve the properties of ACCM [13,14]. These properties include high melting point and hardness, stable chemical properties and strong wear resistance. Moreover, abundant natural resources of Al_2_O_3_ make its industrial production cost effective. Therefore, the prepared ACCM exhibits high strength and good wear resistance with electrical and thermal conductivity quite comparable to that of pure copper.

There are various methods to prepare ACCM, which employs Al_2_O_3_ particles externally or by in situ generation. The methods which involve the external addition of Al_2_O_3_ particles include powder metallurgy [15], mechanical alloying [16], mixing casting [17], and pressureless penetration [18], and so on. The methods involving the in situ generation of reinforcing Al_2_O_3_ particles include internal oxidation [19], self-spreading high-temperature synthesis [20], chemical wrapping [21], sol-gel [22], and so on. Compared with the method of external addition of Al_2_O_3_ particles, the internal in situ generation method has exhibited the several advantages. Firstly, Al_2_O_3_ particles generated during the in situ reaction process are ultra-fine or nano sized, which can effectively improve the binding property between Al_2_O_3_ particles and Cu matrix [23]. Secondly, Al_2_O_3_ particles are synthesized in Cu matrix, which can render an even distribution of these particles in Cu matrix [24]. However, at present, the mainstream internal generation method is complex and the production cost is high. Therefore, the development of a new in-situ reaction method is highly desirable to improve the performance of ACCM.

Recently, solution combustion synthesis (SCS) is introduced as a new in situ wet chemistry synthesis method [25,26,27]. It has the following characteristics. Firstly, a large amount of gas is released during the reaction process producing strong dispersion effect to prevent particles agglomeration. Secondly, the combustion heat, released, makes the reaction self-maintained and completed in an instant. This phenomenon avoids component segregation, and can synthesize uniform combustion product powder with high specific surface area and high activity. Thirdly, the process is simple and fast, low cost, and conducive to industrial production. In this work, by combining SCS, hydrogen reduction method, and in situ generation of reinforced Al_2_O_3_ particles, the ACCP was prepared. Then the ACCM with excellent performance was obtained by the SPS technique. Moreover, the effect of Al_2_O_3_ content and SPS temperature on the microstructure and mechanical property of ACCM was studied in detail, This work has proposed a novel idea for the industrial production of copper-based composite materials.

## 2. Experimental Details

### 2.1. Materials

All the starting materials including cupric nitrate (Cu(NO_3_)_2_), urea (CO(NH_2_)_2_), and aluminum nitrate (Al(NO_3_)_3_·9H_2_O) were of analytical grade. These materials were used to prepare ACCP by combining SCS and hydrogen reduction method. In initial solution, the molar ratio of urea to cupric nitrate (U/Cu) was fixed at 2, and the concentration of cupric nitrate was fixed at 0.1 mol. The additive amount of aluminum nitrate was 0.00126, 0.00254, 0.00386, 0.00519, and 0.00656 mol. Five values, converted to the percent contents (wt.%) of Al_2_O_3_ in ACCM composite, are 1 wt.%, 2 wt.%, 3 wt.%, 4 wt.%, and 5 wt.%, respectively.

### 2.2. Synthesis Procedure

The preparation of ACCP was comprised of following two steps. Firstly, the preparation of (Al_2_O_3_ + CuO) precursor was carried out by SCS method according to the procedure employed in our previous documents [28,29]. Subsequently, the (Al_2_O_3_ + CuO) precursor was subjected to the hydrogen reduction reaction in a flowing H_2_ at 500 °C for 1 h in a tube furnace to prepare ACCP. The flow rate, employed for hydrogen gas, was 0.5 L/min. Finally, the ACCP was put into the graphite mold and performed the SPS process in vacuum environment to obtain ACCM. The heating rate was 100 °C/min with the vacuum was below 0.01 Pa. The sintering pressure was 30 MPa, and the soaking time was 5 min. The effect of various SPS temperatures (675 °C, 725 °C, 775 °C, 825 °C, 875 °C) on the properties of the ACCM was studied in detail.

### 2.3. Characterization

The specific surface area (SSA) of simple was determined by the BET method by using an Automated Surface Area & Pore Size Analyzer (QUADRASORB SI-MP, Quantachrome Instruments, Florida, and American). X-ray diffraction study of samples was carried out in an X-ray diffractometer using CuKa radiation (XRD, D/max-RB12, Rigaku, Tokyo, Japan). The morphology and particle size of samples were observed by a scanning electron microscope (SEM, JSM-6380LV, JEOL, Wuhan, China) with an energy dispersive X-ray spectrometer (EDS, OXFORD Link-ISIS-300, JEOL, Wuhan, China). The relative density and hardness of five SPS samples were measured by the Archimedes method and Vickers hardness tester (HV-30, Shanghai Daling Optical Instrument Co., LTD., Shanghai, China), respectively. The Vickers hardness was performed with a load of 49.0 N force for 20 s and taking an average over five separate measurements. The electrical conductivity of five SPS samples was tested by digital eddy current metal conductor (Sigma2008, Beijing Staavo Technology Co., LTD., Beijing, China), during testing, it needs to pay attention to the parallel of upper and lower end surface, after testing 12 values, remove the maximum and minimum values and then take the average value as the final electrical conductivity value of this sample. The wear resistance of six SPS samples with a size of 2 mm × 10 mm × 10 mm was tested by the high-speed reciprocating friction wear testing machine with a ball-on-disc tribometer (HSR-2M, Lanzhou Zhongke Kaihua Technology Development Co., LTD., Lanzhou, China). The working condition of friction and wear is: a GCr15 steel grinding ball with a diameter of 4 mm and a hardness of 60 HRC, load of 10 N, rotating speed of 500 r/min, friction time of 20 min, and single stroke of 5 mm, test temperature of 23 °C and humidity of 30%.

## 3. Results and Discussion

### 3.1. Preparation of Al_2_O_3_/Cu Composite Powder

ACCP was prepared by combining the SCS and hydrogen reduction methods. Figure 1 depicts XRD pattern and EDS results of the precursor (a,c) and the reduction products (b,d). Only the CuO peak is found in Figure 1a and no obvious Al_2_O_3_ diffraction peak appears due to the small Al_2_O_3_ contents (1 wt.%) or the possibility of Al_2_O_3_ in an amorphous state in the precursor [19]. Moreover, the EDS result (Figure 1c) confirms that the precursor powders only consist of Cu, Al, and O elements. It implies that the precursor is comprised of alumina with cupric oxide (Al_2_O_3_ + CuO). Subsequently, the (Al_2_O_3_ + CuO) precursor is subjected to the hydrogen reduction reaction to prepare ACCP according to procedure. as mentioned in Section 2.2. Figure 1b,d reveals the XRD and EDS results of the reduction products. The absence of CuO and Al_2_O_3_ diffraction peaks in Figure 1b indicates the complete transformation of CuO into Cu. Furthermore, the EDS result (Figure 1d) confirms that the reduction product only consists of Cu, Al, and O elements. Therefore, it is deduced that the absence of Al_2_O_3_ in Figure 1b is still ascribed to the small Al_2_O_3_ content or the possibility of Al_2_O_3_ in an amorphous state in the precursor [19].

Figure 2 shows SEM images of ACCP with various Al_2_O_3_ contents. Compared with the sample without adding Al_2_O_3_ (Figure 2a), it is obvious that other samples containing various Al_2_O_3_ contents exhibit finer particles and more uniform distribution of particle size, which is accordant with their SSA values (Table 1). It is obvious that the SSA of the sample without adding Al_2_O_3_ is the lowest (3.6 m^2^/g). This is because the wettability of Al_2_O_3_ with the Cu matrix is very poor [30], which hinders its effective binding with Cu particles. However, The particle size of the sample with 3 wt.% Al_2_O_3_ content (Figure 2d) is significantly smaller than that of other samples, indicating that dispersion of this sample is the best among all the samples. It is obvious that the sample with 3 wt.% Al_2_O_3_ content is the highest (6.2 m^2^/g) among six samples. With the Al_2_O_3_ content beyond 3 wt.% (such as 4 wt.% and 5 wt.%), the particle size of these samples increases (Figure 2e,f), while the dispersion of these samples decrease caused by the increase of agglomeration of Al_2_O_3_ particles, which is also agree to their SSA trend (see Table 1). This phenomenon also adversely effects the dispersion of Cu particles resulting in their enhanced agglomeration.

### 3.2. Effect of SPS Temperature on the Density and Hardness of ACCM

It is always difficult to fabricate high density ACCM by pressureless sintering technique due to much difference in the melting points of Al_2_O_3_ and Cu. Moreover, grain growth will also occur during liquid phase sintering at high temperature, which will deteriorate the performance of ACCM. The SPS is a new and efficient sintering method which has the various advantages over other techniques. These include fast heating/cooling rates, short sintering time, high density, fine grain growth, short preparation cycle, energy saving, and so on [31,32]. Therefore, the SPS method is utilized to prepare ACCM in the present work.

Figure 3 shows the relative density and hardness of ACCM with 3 wt.% Al_2_O_3_ prepared at various SPS temperatures. It is obvious that with the increase of SPS temperature from 675 °C to 775 °C, the relative density of ACCM increases from 82.49% to 98.19%. At lower SPS temperatures (such as 675 °C), the diffusion rate and atomic movement are slow. With the increase of SPS temperature, the sintering neck grows faster interatomic distance decreases resulting in an increase in the relative density of ACCM. However, the relative density of ACCM decreases slightly with the constant increase of SPS temperatures (such as 775 °C to 875 °C). The higher SPS temperaturse result in the growth of grains (see Figure 4), which reduces overall the relative density of the material. Moreover, at higher SPS temperatures such as 775 °C, the gas in the sintering mass void expands at high temperatures, which also deteriorates the density of ACCM.

It is also obvious in Figure 3 that the hardness of ACCM first increases and then decreases with the increase of SPS temperature. The hardness value reaches at the highest at 775 °C (121.4 HV). Such behavior occurs because with the increase of SPS temperature upto 775 °C, the internal gap defects of ACCM decreases and the relative density of ACCM increases, resulting an increase in the hardness of ACCM. With the SPS temperature continues to rise above 775 °C (such as 825 °C), the hardness of ACCM decreases due to grain coarsening effect.

Figure 4 shows the fracture morphology of ACCM with 3 wt.% Al_2_O_3_ contents at various SPS temperatures. It is obvious that at lower SPS temperature of 675 °C, the sample exhibits a flat fracture surface with the grains exhibiting a strong cubic effect. The sample also exhibits a very clear grain boundary with hole at the fracture surface due to the low sintering temperature and short sintering time [33]. However, with SPS temperature reaching at 725 °C, the grain boundary becomes faded with less number of holes at grain boundary. This sample exhibits the highest value of relative density (see Figure 3). At higher SPS temperature range of 775 °C to 875 °C, a large number of dimples appear at the fracture surface, indicating the strong deformation at fracture surface and the weak bonding among grains in the ACCM composite.

### 3.3. Effect of Al_2_O_3_ Content on the Relative Density and Hardness of ACCM

Figure 5 shows the relative density and hardness of ACCM with various Al_2_O_3_ contents prepared by SPS method at 775 °C. With the Al_2_O_3_ contents increasing from 0 wt.% to 5 wt.%, the relative density of ACCM decreases from 99.89% to 95.32%. This phenomenon can be ascribed to various reasons. Firstly, the presence of higher Al_2_O_3_ contents will reduce the compressibility of Cu particles resulting in the lowering of the density [33]. Secondly, the diffusion of Cu particles is hindered by the distribution of Al_2_O_3_ particles at the Cu crystal boundary, thus further deteriorating the compressibility of the ACCM [34]. Thirdly, the sinterability is also deteriorated by the of increase in interface area of ACCM caused by the addition of Al_2_O_3_ contents.

The hardness of ACCM first increases with the increase of Al_2_O_3_ contents and then decreases. With the Al_2_O_3_ contents increases from 0 wt.% to 3 wt.%, the hardness of ACCM increases from 78.3 HV to 121.4 HV. However, with the Al_2_O_3_ contents above 3 wt.%, the hardness of ACCM slightly decreases from 121.4 HV (3 wt.%) to 106.4 HV (5 wt.%) with the increasing Al_2_O_3_ contents. It is worth noting that the hardness of ACCM with various Al_2_O_3_ contents is still higher than that of the sintered pure Cu sample (0 wt.%), indicating that the increase of Al_2_O_3_ particles can significantly improve the hardness of ACCM. This phenomenon can be ascribed to following reasons. Firstly, Al_2_O_3_ exists as fine particle reinforcement in ACCM, reducing the expansion of the strain zone inside ACCM, and hence improving the ability of ACCM to resist plastic deformation [35]. Secondly, overall hardness of ACCM improves due to the higher hardness of Al_2_O_3_ than that of Cu matrix. Thirdly, the addition of Al_2_O_3_ increases the crystal boundary and the hardness of ACCM by preventing Cu grains from growing up [36]. Fourthly, the addition of Al_2_O_3_ beyond 3 wt.% reduces the density of composite by increasing voids and defects in it. This reduction in density leads to slight decrease in ACCM hardness with the continuous increase in Al_2_O_3_ contents.

According to the sintering theory, the sintering process involves various mechanisms such as gas overflow in the block, the reduction of pores, the formation of the sintering neck and the process of interparticle locking. Figure 6 shows the fracture morphology of ACCM prepared with various Al_2_O_3_ contents. With the Al_2_O_3_ contents in the range of 0 wt.% to 3 wt.%, the fracture mode of ACCM is ductile rupture with the formation of the dimples due to their good plasticity [37]. With the Al_2_O_3_ contents in the range of 4–5 wt.%, the fracture mode is mainly ductile rupture with the appearance of crystal fracture phenomenon. It is also worth noting that with the Al_2_O_3_ contents ranging from 0 wt.% to 2 wt.%, the dimple size of ACCM is large and evenly distributed (Figure 6a–c). With the Al_2_O_3_ contents in the range of 3–5 wt.%, the number of the dimples of ACCM significantly reduces, and the average size of dimple is small and all dimples are evenly distributed (Figure 6d–f). In addition, there are some holes appear in the ACCM. The concave and convex shaped fracture appears, with cracks on it. However, the relatively clean interface between Al_2_O_3_ particles and the Cu matrix exhibit some gaps at the interface, indicating that the binding strength between Al_2_O_3_ particles and copper matrix is poor. Moreover, it also indicates that ACCM with Al_2_O_3_ contents in the range of 1–3 wt.% exhibits higher strength than that containing higher Al_2_O_3_ contents such as 4 wt.% and 5 wt.%. Moreover, it can be seen clearly that Al_2_O_3_ particles in samples with higher Al_2_O_3_ contents (such as 4 wt.% or 5 wt.%) agglomerate at the crystal boundary resulting in weak interface between Al_2_O_3_ and Cu. Therefore, it is concluded that with higher Al_2_O_3_ contents such as 4 wt.% or 5 wt.%, the Al_2_O_3_ is prone to agglomeration, which decreases the density of ACCM, and renders the plastic deformation capacity of ACCM poor.

### 3.4. Effect of Al_2_O_3_ Content on the Electrical Conductivity of ACCM

Figure 7 shows the electrical conductivity of ACCM with various Al_2_O_3_ contents. It is obvious from Figure 7 that with the increase of Al_2_O_3_ contents from 0 wt.% to 5 wt.%, the electrical conductivity of ACCM decreases from 98.24% IACS to 56.66% IACS. However, 80% of total electrical conductivity of ACCM is achieved with Al_2_O_3_ additions up to 3 wt.%, which meets the index requirements of high conductivity and high strength material. Moreover, with the increase of Al_2_O_3_ contents beyond 3 wt.% (such as 4 wt.% or 5 wt.%), the electrical conductivity of ACCM decreases rapidly. Therefore, the optimum reinforcement amount of Al_2_O_3_ lies in the range of 0 wt.% to 3 wt.% in the present work.

There are various advantages of using Al_2_O_3_ as reinforcement in Cu matrix composites. Firstly, Al_2_O_3_ is a nonconducting ceramic, its effect on scattering electrons enhances with the increase in its concentration [38]. Secondly, the addition of Al_2_O_3_ limits the diffusion of Cu by causing dispersion strengthening in ACCM. This phenomenon increases the crystal boundary by refining Cu grains, and hence improves the effect of crystal boundary on scattering during electron transmission [39]. Thirdly, the large internal stress will occur during the SPS process due to mismatch of the thermal expansion coefficients of Al_2_O_3_ and Cu, which further deteriorates the density of ACCM and hence decreases its electrical conductivity.

### 3.5. Effect of Al_2_O_3_ Content on the Friction and Wear Performance of ACCM

Figure 8 shows a relation between friction coefficient and time period for ACCM prepared with various Al_2_O_3_ contents. It is obvious that the friction coefficient of ACCM increases with the increase of Al_2_O_3_ contents. However, with the Al_2_O_3_ contents in the range of 0–2 wt.%, the fluctuation of the friction coefficient is gentle and the run-in time is short. When the Al_2_O_3_ contents are increased to 3–4 wt.%, the fluctuation time increases to 3 min, and this fluctuation range further increases and the run-in time is extended to 5.2 min with the Al_2_O_3_ contents reaching up to 5 wt.%. There are various reasons which are ascribed to above phenomenon. Firstly, Cu matrix is soft, while Al_2_O_3_ is a wear-resistant material, therefore, the friction coefficient of ACCM increases with the increase of Al_2_O_3_ contents [40]. Secondly, during the early stages of the wear test, Cu matrix is worn and peeled off, resulting in the dispersion of Al_2_O_3_ particles on the worn surface in ACCM leading to their high hardness. Therefore, the whole friction process involves relatively large frictional force, which leads to the relatively high friction coefficient of ACCM. Thirdly, Al_2_O_3_ particles are prone to agglomeration in ACCM with the increase of Al_2_O_3_ contents, thus Al_2_O_3_ will be subjected to exfoliation during friction wear experiment, which leads to the larger fluctuation in the friction coefficient value.

Figure 9 shows polishing scratch depths and wear rates of ACCM prepared with various Al_2_O_3_ contents. It is obvious from Figure 9a that polishing scratch depth first decreases with the increase of Al_2_O_3_ contents from 0 wt.% to 3 wt.%, and arrives at a minimum value of 3 wt.%. It is obvious that the polishing scratch depth increases quickly with the increase in Al_2_O_3_ contents from 3 wt.% to 5 wt.%. Moreover, it is evident in Figure 9a that when the Al_2_O_3_ contents are in the range of 0–1 wt.%, the scratches are relatively gentle. However, with the Al_2_O_3_ contents in the range of 2–5 wt.%, the size of scratches increases, indicating that the furrow appears during friction wear experiment and the wear mechanism is mainly grinding abrasion. It is concluded that ACCM with 3 wt.% Al_2_O_3_ exhibits the minimum wear deformation area under the same conditions.

Figure 9b shows the wear rate of ACCM with various Al_2_O_3_ contents. It is clear that the wear rate of samples decreases from 5.679 × 10^−5^ mm^3^/(N·m) to 2.32 × 10^−5^ mm^3^/(N·m) as Al_2_O_3_ contents are increased from 0 wt.% to 3 wt.%. However, with the Al_2_O_3_ contents exceeding 3 wt.% (such as 4 wt.% to 5 wt.%), the wear rate of ACCM increases quickly from 7.732 × 10^−5^ mm^3^/(N·m) to 9.957 × 10^−5^ mm^3^/(N·m). It is also obvious that the wear resistance of ACCM with the range of 1 wt.% to 3 wt.% is higher than that of pure copper (0 wt.%). Moreover, ACCM with 3 wt.% Al_2_O_3_ exhibits the optimum composition in terms of wear resistance. As the Al_2_O_3_ contents are increased up to 3 wt.%, the ground ball exhibits large number of support dots showing that the sample can bear a large payload, and hence reducing the wear of Cu matrix. This behavior can be ascribed to following reasons. Firstly, Al_2_O_3_ is an excellent wear-resistant material, thus, the increase of Al_2_O_3_ content can improve the wear resistance of ACCM [34]. Secondly, during the friction and wear process, the addition of Al_2_O_3_ particle transforms adhesive wear mechanism to grinding grain wear mechanism, which effectively reduces the overall wear of ACCM. Thirdly, a composite lubrication film is formed by very high activity of Al_2_O_3_ particles on the polishing scratch surface under the effect of the load and friction force. This lubrication film will strengthen the self-lubrication effect and improve the wear resistance [41]. Fourthly, although the addition of Al_2_O_3_ in higher amounts (such as 4 wt.% and 5 wt.%) decreases the hardness and strength by adversely affecting the compaction during the sintering process, however, it increases the wear resistance of ACCM by decreasing the wear rate of ACCM. Fifthly, the addition of more Al_2_O_3_ contents (such as 4 wt.% or 5 wt.%) will facilitate more peeling off during friction and wear, leading to the friction wear mechanism of three-body friction pattern among Al_2_O_3_ particles, grinding ball, and ACCM. This phenomenon leads to the complex frictional behavior resulting in an increase in the wear rate of ACCM. Therefore, it is obvious that the ACCM with 3 wt.% Al_2_O_3_ additions exhibit the best wear resistance under the similar conditions.

Figure 10 exhibits the wear morphology of ACCM with various Al_2_O_3_ contents. It is obvious from Figure 10 that the wear mechanism of two samples with small Al_2_O_3_ contents (such as 0 wt.% and 1 wt.%) is mainly adhesive wear, while the wear mechanism of samples with high Al_2_O_3_ contents (such as 2–5 wt.%) is mainly grinding abrasion. It is obvious from Figure 10a that the friction surface of pure copper sample (0 wt.%) exhibits significant exfoliation. This happens due to large difference in hardness of Cu and steel balls, rendering the friction surface under severe plastic deformation resulting in adhesive wear [42].

It is observed that the wear mechanism of ACCM is mainly adhesive wear with small Al_2_O_3_ contents of 1 wt.% (Figure 10b). Nevertheless, with such small Al_2_O_3_ contents, the wear surface is less peeled off, and some holes appear in the wear surface formed by peeling off some Al_2_O_3_ particles during the wear process. When the Al_2_O_3_ contents are 2 wt.% (Figure 10c), furrows with larger gaps appear at the wear surface. Moreover, a slight sheet peeling phenomenon can also be observed, indicating that the main wear mechanism is grinding grain wear and slight adhesive wear. In Figure 10d clearly indicates that ACCM composite with 3 wt.% Al_2_O_3_ contents exhibits the furrows with small gaps at friction surface. The debris shown in Figure 10d (indicated by arrow) are the abrasive dust formed during friction wear process, and the wear mechanism is main grinding grain wear. Figure 10e clearly indicates that with the Al_2_O_3_ contents of 4 wt.%, some deep plows with cracking phenomenon and peeling effects appear at the friction surface, and the wear mechanism is mainly grinding grain wear and slight adhesive wear. In Figure 10f it is obvious that for the composites with 5 wt.% Al_2_O_3_ contents, a deep furrow forms at the friction surface, and significant exfoliation occurs with some holes form at the middle area of the frictional part. In this case, it is obvious that the wear mechanism is a combination of grinding grain wear and adhesive wear. Above results indicate that with the increase of Al_2_O_3_ contents, the binding of Al_2_O_3_ and Cu matrix weakens. This is because Al_2_O_3_ particles are prone to agglomeration with the increase of Al_2_O_3_ contents, which decreases the density of ACCM and consequently increases the friction coefficient of ACCM by increasing the friction resistance caused by increase in roughness of the friction surface. Above phenomenon can be further explained by various reasons. Firstly, severe plastic deformation occurs at surface due to large difference in the hardness of Cu and steel ball with high hardness, leading to sticky wear and tear. Secondly, the addition of Al_2_O_3_ can act as a particle reinforcement, which improves the shear deformation ability of ACCM through the synergistic action. However, it is found that the increase in strength and hardness of ACCM resulting from the increase of Al_2_O_3_ contents is actually caused by a reduction in the size of debris (Figure 10d–f). This similar phenomenon is also explained by previous documents [43,44]. Thirdly, when the Al_2_O_3_ contents are too high (such as 4 wt.% and 5 wt.%), the Al_2_O_3_ particles are separated and detached from ACCM during wearing process, which makes the Al_2_O_3_ particles directly to lose its wear resistance capability. Therefore, cracks appear at the ACCM surface causing surface peeling phenomenon.

## 4. Conclusions

The ACCP is prepared by combining solution combustion synthesis and hydrogen reduction method. Subsequently, the ACCM is prepared by the SPS method. Moreover, the effect of adding various Al_2_O_3_ contents on particle size and morphology of ACCP is studied. Furthermore, the effects of SPS temperature and Al_2_O_3_ contents on the mechanical property of ACCM have also been studied in detail. The following conclusions are drawn.(1)The samples with the addition of various Al_2_O_3_ contents exhibit better dispersity than the samples without the addition of Al_2_O_3_. The ACCP with 3 wt.% Al_2_O_3_ content exhibits the smallest particle size and the best particle dispersity.(2)The relative density and hardness of the ACCM first increase and then decrease with the increase of the SPS temperature. When the SPS temperature is 775 °C, the relative density and hardness of the ACCM reach the maximum value (98.19% and 121.4 HV).(3)The hardness of the ACCM first increases and then decreases with the increase of Al_2_O_3_ contents, and reach the highest value (121.4 HV) with 3 wt.% Al_2_O_3_ content. Furthermore, the relative density and electrical conductivity of the ACCM decrease with the increase of Al_2_O_3_ contents. However, the friction coefficient of the ACCM increases with the increase of Al_2_O_3_ content. Moreover, the wear-resisting property improves first and then deteriorates with increase in Al_2_O_3_ contents. The ACCM with 3 wt.% Al_2_O_3_ content exhibits the highest wear resistance with the minimum wear rate (2.32 × 10^−5^ mm^3^(N·m)).

## Figures and Tables

**Figure 1 materials-16-04819-f001:**
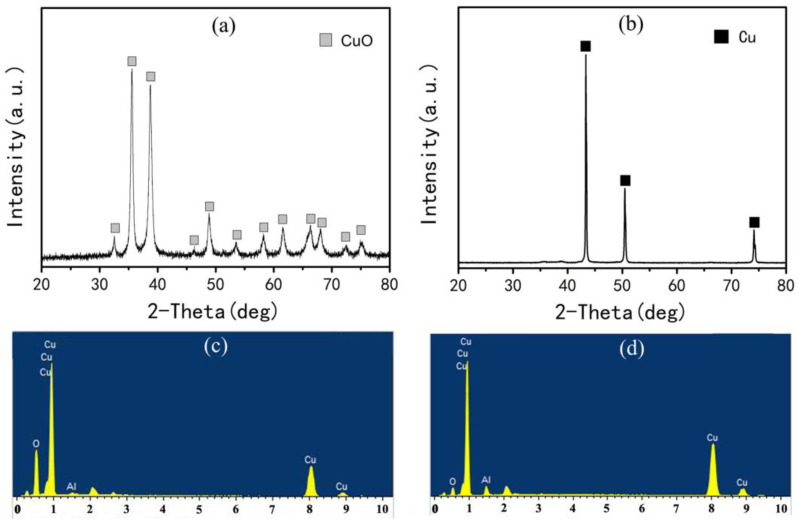
XRD pattern and EDS results of the precursor (**a**,**c**) and the reduction products (**b**,**d**).

**Figure 2 materials-16-04819-f002:**
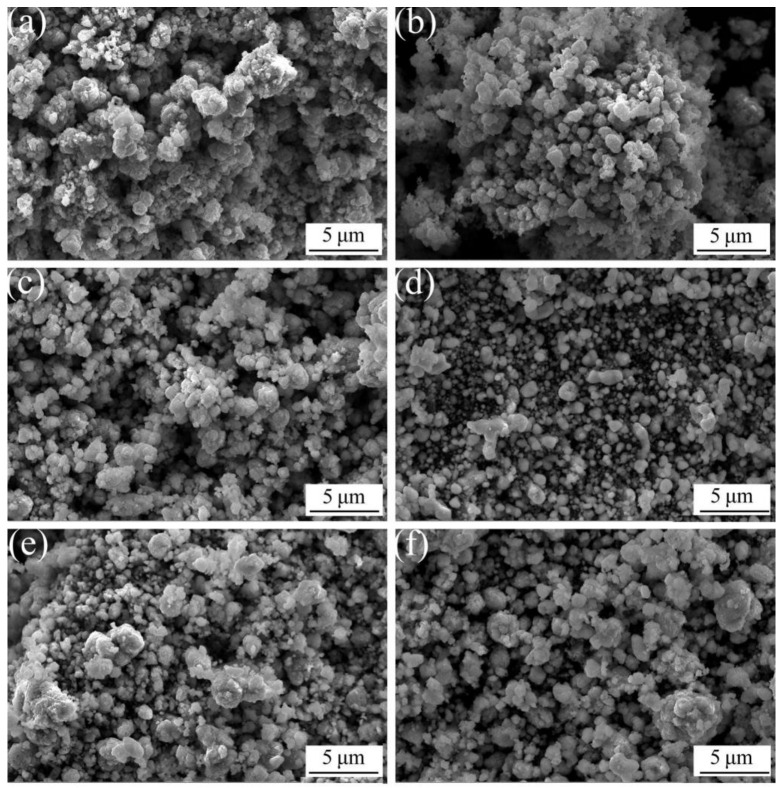
SEM images of ACCP with various Al_2_O_3_ contents: (**a**) 0 wt.%; (**b**) 1 wt.%; (**c**) 2 wt.%; (**d**) 3 wt.%; (**e**) 4 wt.%; (**f**) 5 wt.%.

**Figure 3 materials-16-04819-f003:**
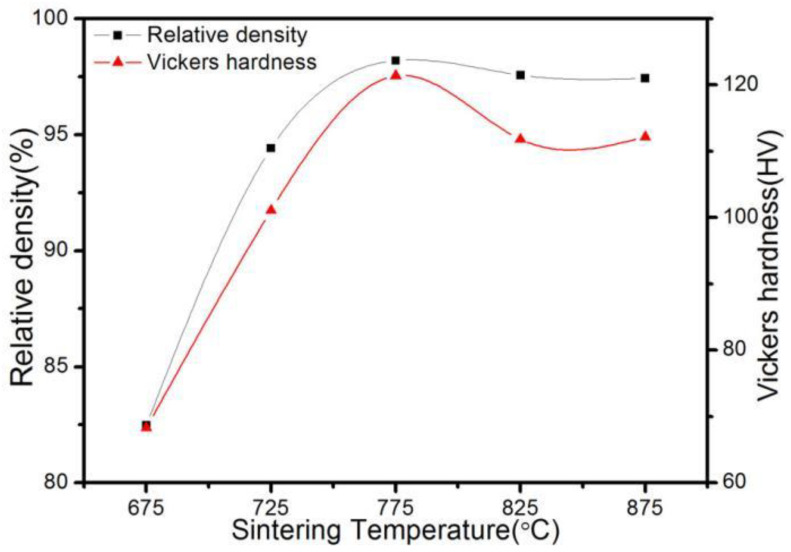
Relative density and hardness of ACCM prepared at various SPS temperatures.

**Figure 4 materials-16-04819-f004:**
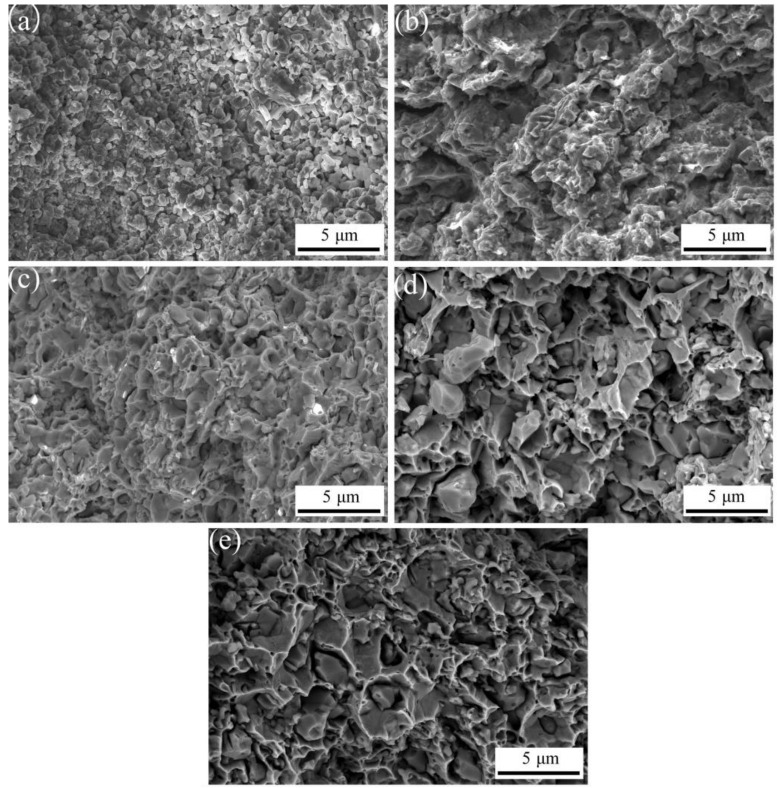
The fracture morphology of 3 wt.% ACCM at various SPS temperatures: (**a**) 675 °C; (**b**) 725 °C; (**c**) 775 °C; (**d**) 825 °C; (**e**) 875 °C.

**Figure 5 materials-16-04819-f005:**
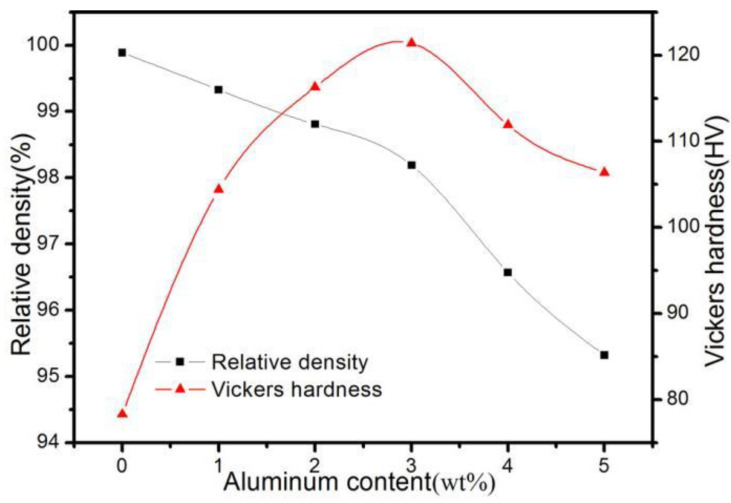
Relative density and hardness of ACCM prepared with various Al_2_O_3_ contents.

**Figure 6 materials-16-04819-f006:**
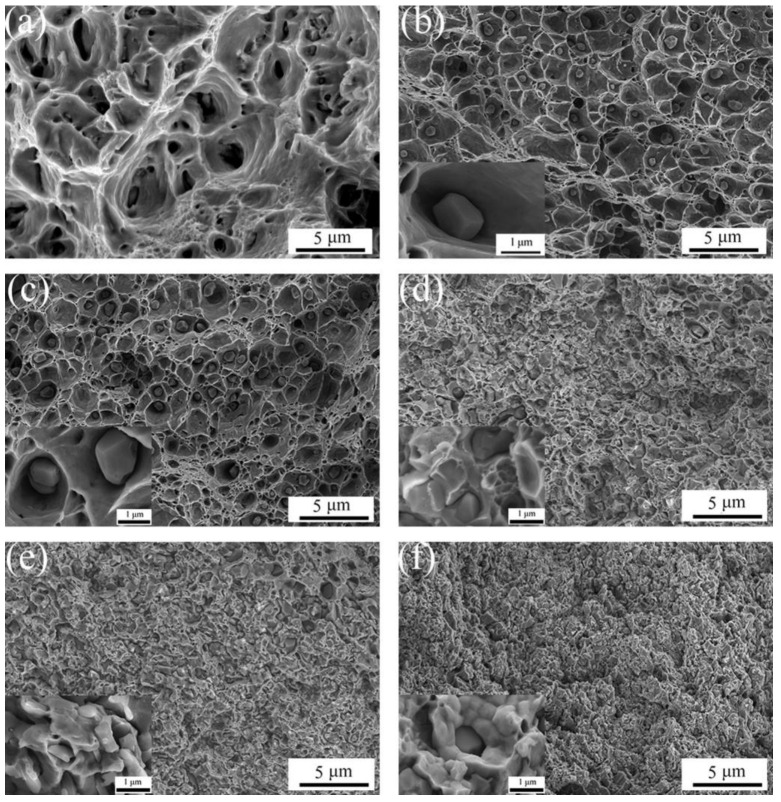
Fracture morphology of ACCM with various Al_2_O_3_ contents: (**a**) 0 wt.%; (**b**) 1 wt.%; (**c**) 2 wt.%; (**d**) 3 wt.%; (**e**) 4 wt.%; (**f**) 5 wt.%.

**Figure 7 materials-16-04819-f007:**
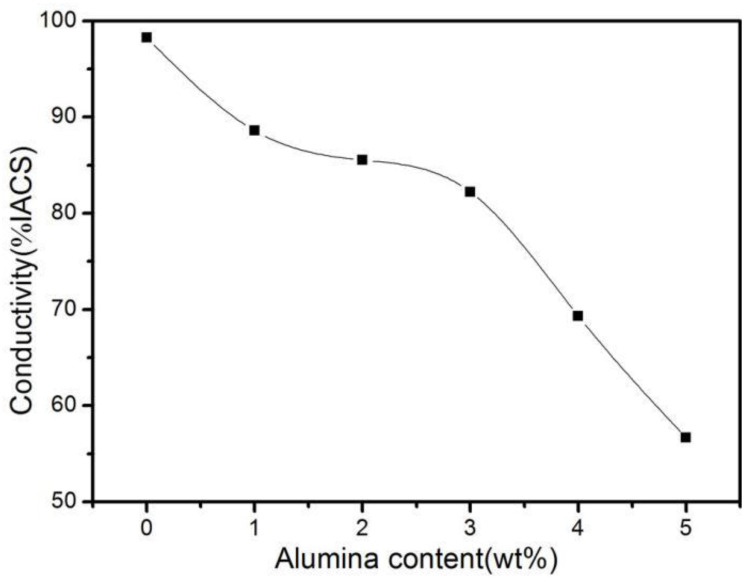
Electrical conductivity of ACCM prepared with various Al_2_O_3_ contents.

**Figure 8 materials-16-04819-f008:**
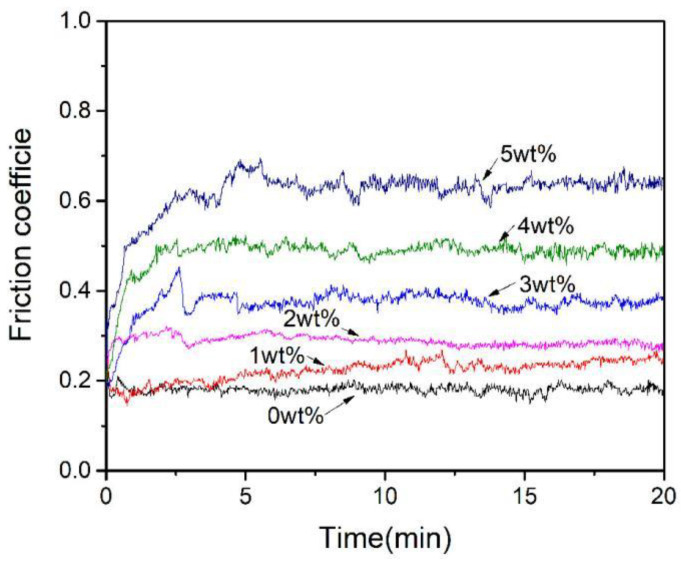
Friction coefficient of ACCM prepared with various Al_2_O_3_ contents.

**Figure 9 materials-16-04819-f009:**
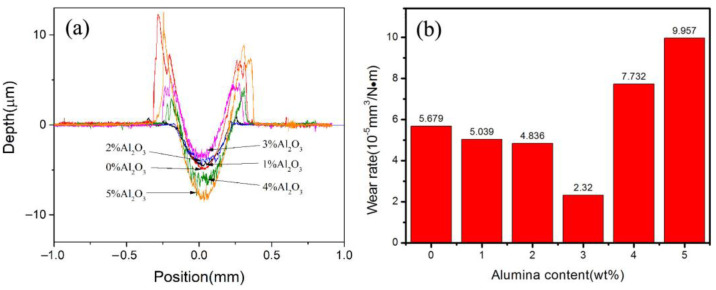
2D contour (**a**) and wear rate (**b**) of ACCM with various Al_2_O_3_ contents.

**Figure 10 materials-16-04819-f010:**
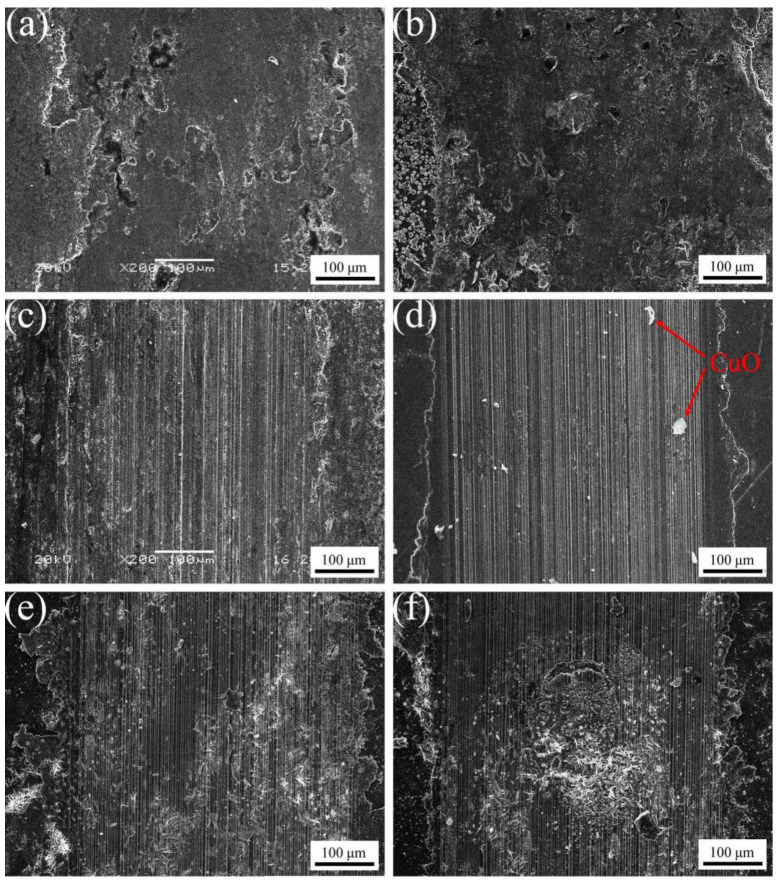
SEM image exhibiting wear morphology of ACCM with various Al_2_O_3_ contents: (**a**) 0 wt.%; (**b**) 1 wt.%; (**c**) 2 wt.%; (**d**) 3 wt.%; (**e**) 4 wt.%; (**f**) 5 wt.%.

**Table 1 materials-16-04819-t001:** SSA of samples containing various Al_2_O_3_ contents.

Al_2_O_3_ content (wt.%)	0	1	2	3	4	5
SSA (m^2^/g)	3.6	4.3	4.8	6.2	5.7	4.5

## Data Availability

Not applicable.

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
