# Peer review of "Preparing and Wear-Resisting Property of Al2O3/Cu Composite Material Enhanced Using Novel In Situ Generated Al2O3 Nanoparticles"

_materials, 2023, doi:10.3390/ma16134819_

Round 1

Reviewer 1 Report

The paper presents the preparing and wear-resisting property of Al2O3/Cu composite material enhanced using novel in situ generated Al2O3 nano-particles. According to the reviewer’s opinion, the paper is well-structured and clear. The topic is interesting and falls within the aim of the journal. In addition, the results are well-presented and could be helpful to further develop the same topic. Therefore, the paper can be accepted for publication in the current form.

the quality of english is ok

Author Response

The paper presents the preparing and wear-resisting property of Al2O3/Cu composite material enhanced using novel in situ generated Al2O3 nano-particles. According to the reviewer’s opinion, the paper is well-structured and clear. The topic is interesting and falls within the aim of the journal. In addition, the results are well-presented and could be helpful to further develop the same topic. Therefore, the paper can be accepted for publication in the current form.

Thanks.

Reviewer 2 Report

The paper entitled “Preparing and wear-resisting property of Al2O3/Cu composite material enhanced using novel in situ generated Al2O3 nano-particles” shows the synthesis of the ACCM by the SPS method. Furthermore, the effects of SPS temperature and Al2O3 contents on the mechanical property of ACCM have been studied.

The scientific quality sounds good and the explanations are clear. The paper deserves to be published after minor revision. You will find here below some points of enhancement:

-        The CMC abbreviation is dedicated to “ceramic matrix composites”,

-        You studied the density and hardness of ACCM for the mechanical properties. Why you didn’t perform tensile test, and you didn’t gauge the Young modulus?

-        What is the method to measure the electrical conductivity? Is it 4-probe method? Please, describe the method.

Author Response

The paper entitled “Preparing and wear-resisting property of Al2O3/Cu composite material enhanced using novel in situ generated Al2O3 nano-particles” shows the synthesis of the ACCM by the SPS method. Furthermore, the effects of SPS temperature and Al2O3 contents on the mechanical property of ACCM have been studied.

The scientific quality sounds good and the explanations are clear. The paper deserves to be published after minor revision. You will find here below some points of enhancement:

- The CMC abbreviation is dedicated to “ceramic matrix composites”,

The abbreviation of Cu matrix composites has been altered to “CuMC” in the revised manuscript now. 

- You studied the density and hardness of ACCM for the mechanical properties. Why you didn’t perform tensile test, and you didn’t gauge the Young modulus?

The test contents for the mechanical property of Cu matrix composites are mainly according to its use. In present work, the use of ACCM is mainly the integrated circuit wire frame and spot welding electrode (mentioned in Introduction), which mainly require outstanding wear resistance and electrical conductivity. However, the density and hardness are closely related to the wear resistance of ACCM, so we mainly studied the density and hardness of ACCM.

- What is the method to measure the electrical conductivity? Is it 4-probe method? Please, describe the method.

Yes, the 4-probe method is used to measure the electrical conductivity in present work. This method has been supplemented in the revised manuscript now.

Reviewer 3 Report

The author investigated the effect of Al2O3 and sintering temperature on the hardness and wear resistance of in situ Al2O3/Cu composite. The presented results are not good enough to be accepted for publication in Materials. I have some comments as below:

1. Abstract should be rewritten.

2. Many important Refs in this field were not cited, so the authors should reorganize the introduction part.

3. The authors mentioned that "the preparation of (Al2O3+CuO) precursor was carried out by SCS method according to the procedure employed in our previous documents [28,29]". However, the Refs No 28 and 29 presented results on other materials (W-Cu, CrN).

4. The characterization part is very rough. How many samples were used for each tests (hardness, wear, density). What kind of counter was used for wear test?, etc,..

5. The authors mentioned the effect of Al2O3 on the dispersion? It is difficult to understand the dispersion of what here?

6. The relationship between the dispersion and particle size should be clarified. Some additional results such as TEM, Zeta size should be conducted and presented.

7. Line 138 to line 144 should be added elsewhere in the Introduction part.

8.  The author mentioned "the density of ACCM increases from 82.49 % to 98.19 %". It is not correct. These value is for relative density as mentioned in Figure 3 and Figure 5. Please correct these mistakes on the discussion  about the relative density of the composite.

9. The author explained the decrease in the density of composited sintered at higher temperatures (825oC, 875oC) is due to the oxidation of grain boundary. This should provide additional result (TEM-EDS, XRD) to support the conclusion.

10. The decrease in hardness of composites sintered at higher sintering temperatures due to the grain growth is correct. However, the supported result (SEM image-Fig. 4) is not good enough. Hardly to identify the grain size with these images. Better metallographic analysis is required in stead of fracture surface of samples.

11.  The friction coefficient of Cu without Al2O3 is below of 0.2. It is difficult to obtain this value by the experimental studies. This value is much lower than theoretical value of Cu (0.39). Has any comments about this?

English should be improved

Author Response

The author investigated the effect of Al2O3 and sintering temperature on the hardness and wear resistance of in situ Al2O3/Cu composite. The presented results are not good enough to be accepted for publication in Materials. I have some comments as below:

  1. Abstract should be rewritten.

The abstract has been rewritten in the revised manuscript now.

  1. Many important Refs in this field were not cited, so the authors should reorganize the introduction part.

Some important Refs in this field have been cited and the introduction part has also been reorganized in the revised manuscript now.

  1. The authors mentioned that "the preparation of (Al2O3+CuO) precursor was carried out by SCS method according to the procedure employed in our previous documents [28,29]". However, the Refs No 28 and 29 presented results on other materials (W-Cu, CrN).

The query of reverent reviewer is very reasonable, here the final prepared materials of the Refs No 28 and 29 are W-Cu and CrN respectively. However, the preparation method (SCS) for the precursor (Al2O3+CuO) is the same with the Refs No 28 and 29. So the synthesis procedure (three of all are SCS method) of precursor for the Refs No 28 and 29 and the present work is the same.

  1. The characterization part is very rough. How many samples were used for each tests (hardness, wear, density). What kind of counter was used for wear test?, etc,..

The characterization part has been rewritten in the revised manuscript now. The ball-on-disc tribometer was used for wear test, which has been added in the revised manuscript now.

  1. 5. The authors mentioned the effect of Al2O3on the dispersion? It is difficult to understand the dispersion of what here?

The query of reverent reviewer is very reasonable. However, here, we are willing to express the effect of Al2O3 on the dispersion that in fact is the effect of Al2O3 on the refinement and distribution of particle size of Cu particles. So the word of dispersion has been explained in the revised manuscript now. 

  1. The relationship between the dispersion and particle size should be clarified. Some additional results such as TEM, Zeta size should be conducted and presented.

The query of reverent reviewer is very reasonable. The BET (specific surface area) values of these powder samples have been conducted and presented in the revised manuscript now.

  1. Line 138 to line 144 should be added elsewhere in the Introduction part.

This may be a misunderstanding among everyone, the content of Line 138 to line 144 is the part of Results and discussion (Fig. 2), which maybe not fit to add to elsewhere in the Introduction part.

  1. The author mentioned "the density of ACCM increases from 82.49 % to 98.19 %". It is not correct. These value is for relative density as mentioned in Figure 3 and Figure 5. Please correct these mistakes on the discussion about the relative density of the composite.

The query of reverent reviewer is very reasonable. After thinking carefully, we also think that there is an inappropriate express for the density of ACCM sample. However, the “density” of ACCM mentioned in present work indicates the “relative density”. So this express wrong has been corrected in the revised manuscript now. 

  1. The author explained the decrease in the density of composited sintered at higher temperatures (825°C, 875°C) is due to the oxidation of grain boundary. This should provide additional result (TEM-EDS, XRD) to support the conclusion.

The query of reverent reviewer is very reasonable. After thinking carefully, we also think that there is an inappropriate express for the oxidation of grain boundary. However, the decrease in the density of composited sintered at higher temperatures (825 °C, 875 °C) is mainly due to the growth of grains. So this express wrong has been corrected in the revised manuscript now.

  1. The decrease in hardness of composites sintered at higher sintering temperatures due to the grain growth is correct. However, the supported result (SEM image-Fig. 4) is not good enough. Hardly to identify the grain size with these images. Better metallographic analysis is required in stead of fracture surface of samples.

I am sorry for delaying to submit this revised manuscript. In fact, those initial samples for measuring fracture morphology have already lost due to the graduation of my postgraduate in early June who administrates those initial samples. Furthermore, our school will enter the summer vacation at this weekend. Therefore, it is very difficult for us to supplement these photos of metallographic analysis in a short time. Moreover, we also think that there is a common view of the grain growth will occur with the increase of sintering temperatures”, it maybe not necessary to prove the phenomenon of grain growth using additional metalgraphic images. Thank you for your comprehending.

  1. The friction coefficient of Cu without Al2O3is below of 0.2. It is difficult to obtain this value by the experimental studies. This value is much lower than theoretical value of Cu (0.39). Has any comments about this?

The query of reverent reviewer is very reasonable. After checking and testifying carefully, we think that the friction coefficient of Cu varies with the types of grinding ball. In present work, the material of grinding ball is GCr15 steel, so this may be the reason why the friction coefficient of pure copper is lower than 0.2.

Round 2

Reviewer 3 Report

Accepted